# A Contribution to the Solid State Forms of Bis(demethoxy)curcumin: Co-Crystal Screening and Characterization

**DOI:** 10.3390/molecules26030720

**Published:** 2021-01-30

**Authors:** Steffi Wünsche, Lina Yuan, Andreas Seidel-Morgenstern, Heike Lorenz

**Affiliations:** 1Max Planck Insitute for Dynamics of Complex Technical Systems, 39106 Magdeburg, Germany; lorenz@mpi-magdeburg.mpg.de; 2Global Drug Development, Novartis, Shanghai 201203, China; lina.yuan@novartis.com; 3Department of Chemical Engineering, Otto von Guericke University, 39106 Magdeburg, Germany; anseidel@ovgu.de

**Keywords:** natural compound, curcuminoids, bis(demethoxy)curcumin, co-crystals, hydroxybenzenes, thermal analysis

## Abstract

Bis(demethoxy)curcumin (BDMC) is one of the main active components found in turmeric. Major drawbacks for its usage are its low aqueous solubility, and the challenging separation from other curcuminoids present in turmeric. Co-crystallization can be applied to alter the physicochemical properties of BDMC in a desired manner. A co-crystal screening of BDMC with four hydroxybenzenes was carried out using four different methods of co-crystal production: crystallization from solution by slow solvent evaporation (SSE), and rapid solvent removal (RSR), liquid-assisted grinding (LAG), and crystallization from the melt phase. Two co-crystal phases of BDMC were obtained with pyrogallol (PYR), and hydroxyquinol (HYQ). PYR-BDMC co-crystals can be obtained only from the melt, while HYQ-BDMC co-crystals could also be produced by LAG. Both co-crystals possess an equimolar composition and reveal an incongruent melting behavior. Infrared spectroscopy demonstrated the presence of BDMC in the diketo form in the PYR co-crystals, while it is in a more stable keto-enol form in the HYQ co-crystals. Solubility measurements in ethanol and an ethanol-water mixture revealed an increase of solubility in the latter, but a slightly negative effect on ethanol solubility. These results are useful for a prospective development of crystallization-based separation processes of chemical similar substances through co-crystallization.

## 1. Introduction

Turmeric (*Curcuma longa L.*) originating in South and Southeast Asia, has been used as a spice and in traditional medicine for centuries, e.g., for the treatment of wounds, colds and arthritic disorders among others [1]. The main bioactive constituents of turmeric are the structurally related curcuminoids curcumin (CUR), demethoxycurcumin (DMC) and bis(demethoxy)curcumin (BDMC), differing only in the presence or absence of one or two methoxy groups (Figure 1). The pharmacological benefits of curcumin, making up 75–80% of the curcuminoids, are the most extensively investigated [2,3,4,5,6,7,8,9,10]. It is reported to possess antioxidative, anti-inflammatory, neuroprotective, antibacterial, antifungal, antineoplastic, and proapopotic activity [3,4,8,9,10], which render it a potential drug candidate for the treatment and prevention of cancer and Alzheimer’s disease [5,7]. Due to its poor bioavailability caused by low aqueous solubility, slow dissolution rate and rapid metabolism it has not yet been approved as a therapeutic [6].

Commercially available CUR usually used for research and clinical trials contains up to 20% of DMC and up to 5% of BDMC [11]. In contrast, it is often believed that CUR alone is accountable for the therapeutic properties of turmeric [2]. For a better understanding of the pharmacological impact of the curcuminoids, it might be favorable to study the therapeutic potential and bioavailability of the individual components because the presence or absence of the methoxy functional group can have a significant impact on binding properties in the human body. Recent studies reveal that CUR might have the highest antioxidant effect but BDMC has higher stability and improved cellular uptake [12]. For example, BDMC can inhibit proliferation and survival of various types of cancer cells (e.g., colon, breast, leukemia) and exhibits the highest anti-metastatic potency in human fibrosarcoma cells [12,13,14].

As the three curcuminoids are chemically quite similar, separation remains challenging. Pure CUR can be obtained from crude curcuminoids by cooling crystallization [15,16]. To obtain pure DMC and BDMC the remaining mother liquor can be treated by liquid chromatography [17]. Another approach is to alter the physicochemical properties of the curcuminoids like the solubility using solvate formation and co-crystallization. Co-crystals are crystalline single phase materials composed of two or more components which are solid at ambient temperatures in a certain stoichiometric ratio [18] whereas solvates include a certain ratio of solvent molecules in their crystal structure [19]. Urbanus et al. demonstrated the potential of co-crystallization as a separation technology by lowering the solubility of a model compound in solution by co-crystallizing it with a suitable coformer to enable a following crystallization step [20]. There are plenty of methods to produce co-crystals, e.g., crystallization from solution, from the melt, or via the solid state like grinding procedures. Co-crystals of CUR are quite well documented, it is reported to form co-crystals, for example, with di- and trihydroxybenzenes [21,22,23,24], nicotinamide [25], isonicotinamide [26], cinnamic acid [27], ascorbic acid [28], and 4,4′-bipyridine-N,N’-dioxide [29], but information about co-crystals of BDMC are sparse. In a patent 6 co-crystals with piperazine, caffeine, L-proline, nicotinamide, isonicotinamide and piperidine are reported but not fully proved [26]. The formation of BDMC solvates was investigated extensively by Yuan et al. [30,31].

This work focuses on the investigation of co-crystal formation of BDMC with hydroxybenzenes. Four methods have been investigated to produce co-crystals: co-crystallization from solution through slow solvent evaporation and rapid solvent removal, co-crystallization via the melt phase, and liquid-assisted grinding. Crystallization from solution via slow solvent evaporation (SSE) and liquid-assisted grinding (LAG) are common techniques for the preparation of stable co-crystals and polymorphs thereof [32,33,34]. Rapid solvent removal (RSR) is a practicable technique for capturing metastable co-crystals [21]. In addition, co-crystallization experiments via the melt phase, e.g., after eutectic melting can be used to investigate the general capability of co-crystal formation of a binary system [35,36]. Since not every technique applied to a co-crystal forming system will result in co-crystal formation due to complex and often unknown kinetics and thermodynamics, all these four techniques have been tested to identify co-crystallizing systems with BDMC. Promising co-crystal candidates were characterized in detail using powder X-ray diffraction (PXRD), differential scanning calorimetry (DSC), thermogravimetry (TGA), Fourier transform infrared spectroscopy (FTIR), optical, and scanning electron microscopy (SEM).

## 2. Results and Discussion

### 2.1. Selection of Coformers

Coformers for the co-crystallization with BDMC were chosen according to the supramolecular synthon approach [37] and due to structural resemblance with CUR [38] in order to conduct a first systematic, yet not exhaustive co-crystal screening. Supramolecular synthons are structural units with functional groups that are connected by intermolecular interactions such as hydrogen bonds and π-π stacking [37]. Through the identification of robust supramolecular synthons, it is possible to select suitable coformers for co-crystallization. BDMC can exist in the β-diketone and the keto- enol form (Figure 2a) whereas only the keto-enol form can be found in crystal structures due to its higher stability [31]. With its hydroxyl, carbonyl and phenyl functional groups the following supramolecular synthons can be identified (Figure 2b):hydroxyl-hydroxyl homosynthoncarbonyl-hydroxyl heterosynthonπ-π interaction

The second approach for selecting suitable coformers is based on structural resemblance to a well investigated molecule and was first introduced by Springuel et al. [38]. They assume that two molecules with similar chemical structure are likely to form co-crystals with identical coformers. They could prove that four out of 10 coformers forming co-crystals with the active pharmaceutical ingredient (API) piracetam also form co-crystals with the structural similar API levitiracetam [38]. Hydroxybenzenes are small molecules with two or more hydroxyl groups bound to a benzene ring. Since they are proved to form co-crystals with CUR [21,22,23,24], four substances out of this group were chosen for the co-crystal screening with BDMC, namely resorcinol (RES), pyrogallol (PYR), phloroglucinol (PHLO), and hydroxyquinol (HYQ) (Figure 3). 

In total, eleven substances out of four substance groups were screened for co-crystallization with BDMC. Besides hydroxybenzenes, aromatic hydroxy acids (salicylic acid, *p*-hydroxybenzoic acid, *trans*-ferulic acid), an aliphatic hydroxy acid (L-tartaric acid) and dicarboxylic acids (fumaric acid, succinic acid, adipic acid) were tested [39]. Since no co-crystal formation could be observed for the investigated other substance groups, it was focused on co-crystallization with hydroxybenzenes.

### 2.2. Co-Crystallization of Bis(demethoxy)curcumin with Hydroxybenzenes

Different methods to produce co-crystals have been investigated to screen for co-crystals of BDMC and hydroxybenzenes. Ethanol was used for every technique requiring a solvent since it forms no solvate with BDMC and has low toxicity [30]. Results of the screening including DSC thermograms and PXRD patterns are shown in Figure 4 to 6 for resorcinol (RES), pyrogallol (PYR), and hydroxyquinol (HYQ), respectively. 

For resorcinol (RES) (Figure 4), as well as for phloroglucinol (PHLO) (Appendix A), no co-crystal formation could be detected. The DSC measurements (Figure 4a) of all co-crystallization products and the physical mixture look similar and indicate a simple eutectic with a sharp eutectic melting peak followed by a broad melting peak of excessive BDMC. In case of RES as coformer two eutectic melting peaks are apparent since RES forms two polymorphs α and β [40]. The phase transition from form α to form β can be seen from the DSC measurement of the physical mixture at a heating rate of 2 °C·min^−1^ as an exothermic recrystallization event between the two melting peaks. Powder diffractograms (Figure 4b) reveal no new diffraction peaks that could indicate a new solid phase. Additional peaks at 2θ values of 20.9°, 22.5° and 27.7° can be seen in the samples produced by RSR and DSC of a 1:1 physical mixture at 125 °C. They are affiliated to the presence of the β polymorph of RES formed during the experiment whereas the reference substance consists of almost solely polymorph α [41].

Curcumin and RES form stable co-crystals with multiple recognition sites, and strong H-bonding between the hydroxyl groups of RES and the hydroxyl, methoxy and carbonyl groups of CUR together with weaker π···H-C and O···H-C interactions for close packing [22]. BDMC is lacking methoxy groups, which are pivotal for the formation of the CUR-RES co-crystal and shows up with a lower number of recognition points for strong H-bonding. Together with the close packing of the BDMC crystal itself this might be the reason for the impeded co-crystal formation with RES. Chow et al. proved the existence of a metastable co-crystal of CUR and PHLO by rapid solvent removal technique with different solvents [21]. Since PHLO and RES differ in the presence or absence of a hydroxyl group in the C_5_ position, this factor can be regarded as the crucial factor for co-crystal formation. The additional hydroxyl group may impede close packing of CUR and PHLO molecules, which in return implies a weakening of intermolecular interactions. This can also be the reason why no co-crystal of BDMC and PHLO was detected. From these experiments it can be concluded that a thermodynamically stable co-crystal of BDMC and either RES or PHLO does not exist. However, it should be mentioned that the results do not mean that there is no possibility of co-crystal formation at all. Varying the solvents, the amount of solvent and changing the stoichiometry of the binary system might lead to the discovery of kinetically stable co-crystal phases in the future.

From the co-crystallization experiments with PYR (Figure 5) the existence of a co-crystal can be assumed since all DSC measurements (Figure 5a) show a sharp endothermic peak after the eutectic melting. Pure PYR and BDMC have a melting point of 131 °C and 237 °C, respectively, at a heating rate of 10 °C·min^−1^. Eutectic melting occurs below the melting point of pure PYR at temperatures between 122 °C and 126 °C. The second endothermic event starts at temperatures between 164 °C and 167 °C and can be interpreted as the melting of an intermediate solid phase, a co-crystal phase. This assumption could be confirmed by a DSC measurement of a physical mixture of BDMC and PYR in a 1:1 molar ratio at a heating rate of 2 °C·min^−1^. After eutectic melting starting at 126 °C, an exothermic peak can be observed indicating the formation and recrystallization of a co-crystal phase from the eutectic melt. The small endothermic event at 93 °C in the thermogram of the SSE sample can be assigned to residual solvent EtOH. In addition, there is a high probability that PYR underwent hydrate formation with water molecules dissolved in the solvent because PYR forms a stable 0.25 hydrate [42]. In this sample, the formation and recrystallization of the co-crystal is also indicated by an exothermic event after eutectic melting. The PXRD patterns in Figure 5b show no distinct reflections different from the starting materials except for the DSC sample heated to 150 °C. Completely new diffraction peaks at 2θ values of 12.48°, 15.64°, 19.65°, 21.01°, and 24.40° are present. This suggests that the co-crystal did not form at ambient temperatures and the formation of the co-crystal is coupled to the precondition of eutectic melting or the application of thermal energy. Hence, thermal analysis is a suitable method for the detection of co-crystal forming systems, even in cases where other methods are not successful, e.g., because the co-crystal forming region is completely missed due to large differences in the solubility of the compounds [36]. In addition, it requires only small amounts of substances and there is no need for a solvent, which is favorable with respect to green chemistry. In contrast to BDMC, curcumin forms stable co-crystals with PYR at ambient temperatures [22]. Again, the CUR methoxy group plays a vital role for interconnecting the molecules. From the experiments carried out within this work, a co-crystal phase of BDMC and PYR only formed at ~125 °C, a temperature above the eutectic melting.

DSC measurements of the co-crystallization experiments of BDMC with hydroxyquinol (HYQ) (Figure 6a) confirm co-crystal formation for all experiments through a sharp endothermic peak at a temperature of 179 °C to 192 °C. Thermal analysis of the 1:1 physical mixture demonstrates the formation of the co-crystal after eutectic melting which is indicated by an instantly following exothermic peak as aforementioned above for the BDMC-PYR system. This recrystallization of the co-crystal is also observed for the RSR and SSE samples. The DSC curve of the product from the liquid-assisted grinding (LAG) experiment is missing the eutectic melting peak. Thus, it can be concluded that the co-crystal not only forms upon heating but also by grinding. It can also be expected that this co-crystal is the thermodynamically favored phase. Since a broad endothermic peak suggesting melting of excessive BDMC can be observed after the co-crystal melting in the DSC curve of the LAG sample, the co-crystal composition was proved by co-crystallization experiments with different compositions of BDMC and HYQ (see Section 2.4).

The PXRD pattern of the LAG sample, and the sample produced by DSC at 160 °C (Figure 6b) give clear evidence that a new co-crystal phase was formed. There are new diffraction peaks at 2θ values of 7.64°, 17.97°, 18.83°, 24.20°, 24.67°, 28.51°, and 30.43° while most of the characteristic peaks of pure BDMC and HYQ are missing. Co-crystallization experiments from the solution (SSE and RSR) did not yield in the formation of a co-crystal. It is quite likely that the large difference in solubility of BDMC and HYQ in the used solvent ethanol leads to the failure in co-crystal formation from solution.

### 2.3. Characterization of Pyrogallol-Bis(demethoxy)curcumin Co-Crystal

#### 2.3.1. Composition of the PYR-BDMC Co-Crystal

To confirm the co-crystal composition of the PYR-BDMC co-crystal, co-crystallization experiments from the melt phase with varying composition of BDMC and PYR were carried out using DSC by heating and annealing the solid mixtures to 150 °C. Afterwards, the samples with varying molar composition from $x_{\mathrm{PYR}}$ = 0.40 to 0.70 were characterized by DSC (Figure 7). From these thermograms, a co-crystal composition with $x_{\mathrm{PYR}}$ higher than 0.50 can be excluded. Tiny endothermic events at temperatures close to the eutectic temperature of the system (122 °C–126 °C) indicate melting of excessive PYR, and its 0.25 hydrate that formed upon the storage of PYR. The highest melting enthalpy is recognized for the 1:1 molar composition with 86.4 J·g^−1^. The higher melting enthalpy indicates a higher purity of the compound and a lower amount of impurities or unreacted starting material. Thus, the co-crystal composition can be concluded to be equimolar with a co-crystal melting point of 164 °C.

After co-crystal melting a broad endothermic shoulder is visible for all compositions indicating an incongruent melting behavior. Compounds with an incongruent melting behavior melt under decomposition into a liquid phase with a composition different from the initial solid phase, and a solid phase which melts gradually afterwards. The incongruent melting temperature of the co-crystal corresponds to the peritectic invariant $T_{p}$ in the binary PYR-BDMC system. 

#### 2.3.2. Thermogravimetric Analysis of the PYR-BDMC Co-Crystal and its Single Components

The thermal behavior and stability of the co-crystal and its constituents were investigated by thermogravimetric analyses coupled with DSC (TGA-DSC) (Figure 8). In addition, a 1:1 physical mixture of PYR and BDMC was analyzed. Unlike DSC, TGA-DSC measurements were carried out in open crucibles.

Pure BDMC (Figure 8a) starts to lose mass right from the beginning of the measurement. Thermal stability investigations of the three curcuminoids by Heffernan et al. revealed that BDMC shows the lowest thermal stability among the curcuminoids [17]. On the other hand, it is thermodynamically more stable due to its high melting point [43]. Before the melting starts at 232 °C, a mass loss of 1.66% is recognized. Another reason for this mass loss could be partial formation of a BDMC hydrate upon storage. The existence of a monohydrate was already confirmed by Karlsen et al. [44]. This assumption is supported by the presence of a broad endothermic effect starting at ~120 °C, which was not detectable by DSC measurements conducted in a closed crucible. Already during melting degradation of BDMC occurred, indicated by the exothermic peak upon melting as well as the rapid ongoing loss of mass. Until the final temperature of 270 °C, the total mass loss was 8.84%. As the other samples are heated to 240 °C and for the reason of comparability, the mass loss of BDMC until that temperature was also calculated to 3.35%.

Pure PYR (Figure 8b) shows a small endothermic peak at ~80 °C that corresponds to the dehydration of the formed 0.25 hydrate upon storage. The weight loss due to dehydration adds up to 2.74% which equals 0.20 mol per mol PYR. Melting of PYR starts at 135 °C and is instantly followed by almost complete evaporation far below the boiling point of 309 °C [45]. Beyond its melting temperature PYR seems to be very volatile at ambient pressure. 

From the thermogram of the 1:1 physical mixture of PYR and BDMC (Figure 8c) eutectic melting at 130 °C followed by an exothermic peak signaling co-crystal formation is clearly visible. Losses in sample mass already occur at temperatures below eutectic melting and can be assigned to mainly nascent decomposition of BDMC. Furthermore, after eutectic melting PYR molecules can be transferred into the gas phase. The overall mass loss at 240 °C reaches 25.94%. The total weight loss of the co-crystal produced by DSC (Figure 8d) is even lower and counts to 23.33%. The theoretical mass loss of a 1:1 molar composition of PYR and BDMC, calculated from the mass loss of its single components, would count to almost 30%. Thus, these measurements lead to the conclusion that the PYR-BDMC co-crystal shows a higher thermal stability or a less volatility of about 20% than its single constituents.

#### 2.3.3. Fourier Transform Infrared Spectroscopy (FTIR) of the PYR-BDMC Co-Crystal and its Single Components

The FTIR spectra of PYR, BDMC, and PYR-BDMC co-crystal are shown in Figure 9. Infrared spectroscopy (IR) provides valuable information about the vibrational modes of a substance. By comparing the IR spectrum of the PYR-BDMC co-crystal with its single constituents, it is possible to detect changes in hydrogen bonding and molecular conformations [46]. A comparison of IR vibration modes is summarized in Table 1. Phenolic hydroxyl groups absorb strongly between 3705 and 3125 cm^−1^ and show broad peaks for hydrogen bonded OH groups while free OH groups reveal a sharp peak. The OH stretching frequencies of PYR are located at 3418, 3370, and 3230 cm^−1^, BDMC absorbs at 3491, and 3214 cm^−1^. The absorption peaks of the co-crystal differ from that of the single constituents giving evidence for the change in the chemical environment of the phenolic OH groups. Interestingly, the peak at 3474 cm^−1^ is sharp. It indicates a dramatic weakening of hydrogen bonding of one of the OH groups or the presence of a free OH group that is not involved in building up the crystal structure. The other two peaks at 3350 and 3215 cm^−1^ shifted to lower frequencies suggesting that they are involved in strong intermolecular hydrogen bonding within the co-crystal phase.

Two additional peaks in the co-crystal spectrum at 2922 and 2851 cm^−1^ imply the presence of a methylene group. This group is exclusively found in the diketo form of BDMC, so it may be concluded that this is the molecular structure of BDMC present in the co-crystal. Since BDMC occurs mainly in the keto-enol form in the solvent EtOH and the diketo form is the less stable form of BDMC, these may be the reasons for the inefficient production of PYR-BDMC co-crystals except of crystallization from the eutectic melt. In addition, the C=O stretching peak of the co-crystal at 1623 cm^−1^ has higher intensity than that of BDMC, while the enol C-O peak at 1429 cm^−1^ has a distinctly lower intensity.

#### 2.3.4. Microscopic Analysis

Images of the PYR-BDMC co-crystal crystallized from the eutectic melt were recorded using scanning electron microscopy (SEM) (Figure 10). The starting materials BDMC (Figure 10a), and PYR (Figure 10b) both crystallize in a monoclinic crystal system [20,35]. BDMC exhibits a prismatic crystal habit, while PYR crystals are more needle-like. The co-crystal crystallized from the melt phase (Figure 10c,d) demonstrates no distinct crystal habit. Particles are very small in size and are greatly shaped in an irregular manner.

### 2.4. Characterization of Hydroxyquinol-Bis(demethoxy)curcumin Co-Crystal

#### 2.4.1. Composition of the HYQ-BDMC Co-Crystal

As for the PYR-BDMC co-crystal, the composition of the HYQ-BDMC co-crystal was confirmed by DSC measurements of samples with varying compositions of HYQ, and BDMC from $x_{\mathrm{HYQ}}$ = 0.40 to 0.60 (Figure 11). The samples were prepared by liquid-assisted grinding (LAG) for 90 min. At molar ratios of $x_{\mathrm{HYQ}}$ higher than 0.50 an endothermic event close to the eutectic melting point of the binary system (135–139 °C, see Figure 6a) was detected indicating an excessive amount of HYQ within the system. For $x_{\mathrm{HYQ}}$ ≤ 0.50 the highest co-crystal melting enthalpy is observed for $x_{\mathrm{HYQ}}$ = 0.50 with $\Delta H_{F}$ = 115.2 J·g^−1^. Hence, the molar co-crystal composition is supposed to be 1:1. For an explicit clarification of the co-crystal composition, single crystal X-ray diffraction (SCXRD) for structural elucidation is required. But to date, the growth of single crystals suitable for SCXRD was not successful despite testing various solvents (ethanol, acetone, acetonitrile, toluene, ethyl acetate, chloroform), solvent mixtures (toluene/ethanol 80/20 *v*/*v*, toluene/ethyl acetate 50/50 *v*/*v*, toluene/ethyl acetate 90/10 *v*/*v*) and compositions of HYQ, and BDMC.

As well as the PYR-BDMC co-crystal, the HYQ-BDMC co-crystal reveals an incongruent melting behavior, which is indicated by the broad endothermic event beyond co-crystal melting at the peritectic temperature $T_{p}$ of 180 °C.

An incongruent melting behavior can often be observed for compounds with weak intermolecular interactions, or low thermal stability [35]. It is also typical for binary systems with a large gap between the melting points of the individual substances as in this case. BDMC has a high melting point at 232.8 °C, whereas PYR and HYQ melt at temperatures of 134.4 °C and 142.3 °C, respectively. For curcumin-hydroxybenzene co-crystals only a congruent melting behavior is reported in the literature [24]. Thus, it can be considered that the lower thermal stability of BDMC and the gap between the melting points of the pure substances might be the main reasons for the incongruent melting behavior of BDMC co-crystals.

#### 2.4.2. Thermogravimetric Analysis of the HYQ-BDMC Co-Crystal and its Single Components

Thermograms of the TGA-DSC measurements of a 1:1 physical mixture, and the HYQ-BDMC co-crystal are shown in Figure 12. The analysis of pure HYQ (Appendix A) shows a similar thermal behavior as pure PYR. Melting of HYQ in an open crucible starts at 142 °C. Beyond the melting temperature, a rapid weight loss mainly due to passage into the gas phase can be observed. In contrast to PYR, the overall weight loss of HYQ after heating to 240 °C is much lower and counts to 58.55%. From the thermogram of the 1:1 physical mixture of HYQ and BDMC (Figure 12a) eutectic melting starts at 141 °C. It is followed by an exothermic peak corresponding to co-crystal crystallization. Loss in sample mass already occurs at temperatures below eutectic melting indicating the starting degradation of BDMC. The overall mass loss after heating to 240 °C amounts to 13.26%.

The weight loss of the co-crystal upon heating up to 240 °C (Figure 12b) is with 13.03% quite like that of the physical mixture. But it is lower than the weight loss that can be calculated from the TGA measurements of the single components HYQ and BDMC in the respective ratio which would be about 19%. In summary, it can be stated that the thermal stability of the co-crystal is about 30% higher than that of its single constituents.

#### 2.4.3. Fourier Transform Infrared Spectroscopy of the HYQ-BDMC Co-Crystal and its Single Components

The FTIR spectra of HYQ, BDMC, and the HYQ-BDMC co-crystal are shown in Figure 13, and a comparison of the vibration modes of various functional groups is given in Table 2. Phenolic hydroxyl group stretching of the co-crystal shifted to higher frequencies indicating a weakening of hydrogen bonding. One sharp peak at 3530 cm^−1^ leads to the conclusion that there is one free OH group present in the co-crystal that is not necessarily required for arranging the molecules in the co-crystal lattice. The BDMC OH groups can be regarded as essential for molecule assembly. Thus, probably one OH group in the HYQ molecule is free. It can also be assumed that it is one in the *ortho* position because RES with two OH groups in the *ortho* position did not yield in co-crystal formation so far.

No significant signal shift for C=O stretching was evident suggesting an inactive role of this group in intermolecular interactions within the co-crystal. The absorption peak of the enol C-O group in the BDMC molecule shifted dramatically to lower frequencies for the co-crystal. This implies an increase and strengthening in intermolecular interactions, e.g., hydrogen bonding. As already mentioned, to finally proof these assumptions a distinct clarification of the crystal structure by single crystal XRD is pivotal.

#### 2.4.4. Solubility Measurements

Co-crystallization is often performed to alter the physicochemical properties of a crystalline material in a desired manner, e.g., the solubility. Therefore, solubility measurements of the target compound BDMC bound in the co-crystal with HYQ were conducted in the solvents EtOH and EtOH/H_2_O (50/50 *v*/*v*) and compared to the solubility of pure BDMC (Table 3). The solubility of BDMC in pure EtOH within the co-crystal does not change significantly compared to pure BDMC. The results rather show a slight decrease in solubility from 43.25 to 39.04 g·L^−1^. In this case, the amount of dissolved HYQ molecules may interfere with the dissolution and solvation of BDMC. In the end, this can influence the overall solubility of BDMC in EtOH in a negative way to some degree. Actually, a decrease in solubility due to co-crystal formation can be quite useful for future investigations on curcuminoid separation with the help of co-crystallization.

The solubility of BDMC in EtOH/H_2_O is very low. It can be impressively increased by over 100% to 0.83 g·L^−1^ through co-crystallization with HYQ. The coformer HYQ possesses a high ability to promote the dissolution of BDMC molecules in aqueous EtOH. An increase in solubility of the slightly water-soluble natural compound BDMC is highly desirable with regard to improving the bioavailability. The results of this small-scale investigation on the change in solubility of BDMC due to co-crystallization show the diverse influence on the physicochemical properties of a compound.

#### 2.4.5. Microscopic Analysis

Light microscopic and SEM images of the HYQ-BDMC co-crystal are presented in Figure 14 and Figure 15, respectively. Freshly nucleated co-crystals from an ethyl acetate solution were recorded using polarized light (Figure 14a). Before HYQ-BDMC co-crystals started to nucleate and grow, pure BDMC crystallized as well (Appendix A). Thus, the concentration of HYQ and BDMC in the starting solution was not optimal for only reaching the co-crystal forming region. The co-crystals grow as sherulites from a highly supersaturated solution, which is also visible in S3b-d. A section of a large co-crystal grown from toluene/ethyl acetate (50/50 *v*/*v*) is depicted in Figure 14b. Although very slow solvent evaporation was provided, the co-crystals grew as polycrystalline aggregates.

SEM images of the HYQ-BDMC co-crystals produced by LAG and grown from toluene/ethyl acetate (50/50 *v*/*v*) are shown in Figure 15a,b, respectively. Micrographs of the starting materials are shown in Appendix A. The LAG sample in Figure 15a consists mainly of a very fine powder with primary particle sizes of about 1 µm. Among these small particles, larger plate-like particles can be seen. The co-crystal grown from solution in Figure 15b originates from the sample also used for the light microscope image in Figure 14b. The crystalline particle shown here is built up of several thin layers that are loosely held together. The texture seems to be brittle, and porous as a result of the spherulitic growth.

## 3. Materials and Methods

### 3.1. Materials

Bis(demethoxy)curcumin (BDMC, CAS No. 24939-16-0), with purity higher than 98%, was obtained from TCI (Tokyo, Japan). Resorcinol (RES, CAS No. 108-46-3), pyrogallol (PYR, CAS No. 87-66-1), phloroglucinol (PHLO, CAS No. 108-73-6), hydroxyquinol (HYQ, CAS No. 533-73-3) with purity higher than 98% were purchased from Sigma-Aldrich (Steinheim, Germany). Solvents (ethanol, acetone, acetonitrile, ethyl acetate, toluene, chloroform) of HPLC grade were purchased from commercial suppliers. All materials were used as received without any further purification.

### 3.2. Preparation of BDMC Co-Crystals

#### 3.2.1. Liquid-Assisted Grinding (LAG)

Liquid-assisted grinding experiments were performed on a MM400 mixing mill (Retsch, Haan, Germany) at a vibrational frequency of 30 Hz. Samples of 200 mg containing BDMC and the respective coformer in a 1:1 molar ratio and 40 µL of ethanol together with one stainless steel ball of 10 mm diameter were put into a 10-mL screw-top grinding jar made of stainless steel and were ground for 30 min. After grinding, the jar was left open for solvent evaporation. Then the samples were homogenized in a mortar and stored in a 5-mL glass vial for further analyses. Samples for the confirmation of the HYQ-BDMC co-crystal composition were prepared by LAG with different molar ratios of HYQ ranging from 0.40 to 0.60 and a grinding time of 90 min.

#### 3.2.2. Crystallization from Eutectic Melt

In addition to grinding, the thermal contact method was applied to investigate the phase behavior of binary physical mixtures of BDMC and the respective coformer upon heating. Physical mixtures in a 1:1 molar ratio were prepared by grinding the single components separately in a mortar, weighing them into a 5-mL glass vial and gently mixing them through shaking the vial. Samples of 4.0 ± 0.5 mg were placed into a sealable aluminum crucible for DSC measurement. Every mixture was measured over a temperature range from 25 °C to 270 °C at heating rates of 2 °C·min^−1^ and 10 °C·min^−1^.

The confirmation of the PYR-BDMC co-crystal composition was performed using a modified DSC temperature program as follows: Physical mixtures (7.0 ± 0.5 mg) of BDMC and PYR with molar ratios of PYR ranging from 0.40 to 0.70 were heated to 150 °C at a heating rate of 2 °C·min^−1^ and kept there for 30 min to allow the formation of the co-crystal. After cooling down to room temperature the mixture was heated to 240 °C at a heating rate of 2 °C·min^−1^.

#### 3.2.3. Crystallization from Solution

Crystallization from solution was performed via slow solvent evaporation (SSE) and rapid solvent removal (RSR). For the slow solvent evaporation experiments, samples of 250 mg of BDMC and the respective coformer in a 1:1 molar ratio were placed in a 5-mL glass vial. 3 mL of ethanol were added, and the vials were closed with a cap. The mixture was heated close to the boiling point of ethanol until complete dissolution occurred. The solutions were filtered while hot using a 0.45-µm membrane filter, and the clear solutions were then allowed to cool down to room temperature. The caps of the vials were removed, and the vials were sealed with a perforated parafilm for slow evaporation of ethanol. After complete evaporation of ethanol, the obtained crystals were kept in closed vials for further analyses.

For the rapid solvent removal experiments, samples of 200 mg of BDMC and the respective coformer in a 1:1 molar ratio were placed in a 10-mL flask. 5–7 mL of ethanol were added and the mixture was sonicated until complete dissolution of the solids occurred. Ethanol was then removed using a rotary evaporator under vacuum (130 mbar) and a water bath (50 °C). The resulting product was dried in a vacuum oven at room temperature for 10 h to remove residual ethanol, ground to a fine product and stored in a closed vial for further analyses.

### 3.3. Powder X-ray Diffraction (PXRD)

X-ray powder data were collected on an X’Pert Pro diffractometer (PANalytical GmbH, Kassel, Germany) using Cu-Kα radiation (λ = 1.5418Å). Samples were prepared on a ‘zero-background’ sample holder and scanned in a 2θ range of 3° to 40° with a step size of 0.017° and a counting time of 50 s per step at ambient temperature.

### 3.4. Differential Scanning Calorimetry (DSC)

Differential scanning calorimetry (DSC) analyses were performed on a DSC3 instrument (Mettler Toledo, Gießen, Germany). Samples of 4.0 ± 0.5 mg were put into a 40-µL aluminum crucible, sealed and heated at a rate of 2 °C·min^−1^ or 10 °C·min^−1^. Nitrogen was used as the purge gas at a flow rate of 30 mL·min^−1^.

### 3.5. Thermogravimetric Analysis (TGA)

Thermogravimetric analyses (TGA) were performed on a Sensys Evo TGA-DSC (Calvet-type) instrument (Setaram, Caluire-et-Cuire, France). Samples were put in an open aluminum crucible and heated at a heating rate of 2 °C·min^−1^. Helium was used as the purge gas. Both, the sample mass (TGA) and the heat flow (DSC) profiles were recorded.

### 3.6. Fourier Transform Infrared Spectroscopy (FTIR)

The FTIR spectra of solid samples were obtained in attenuated total reflection mode (ATR) using an Alpha II spectrometer (Bruker, Billerica, MA, USA). Powdered samples were scanned from 4000 to 400 cm^−1^ at an interval of 1.03 cm^−1^.

### 3.7. Solubility Measurements

The gravimetric method was used to determine the solubility of BDMC and the HYQ-BDMC co-crystal produced by LAG in EtOH and EtOH/H_2_O (50/50 *v*/*v*) at room temperature. An excessive amount of solid was mixed with the respective solvent, and the suspension was stirred at 25 °C ± 1 °C for 72 h to reach equilibrium. Before sampling, excess solid was allowed to settle. Saturated solutions were filtered through a 0.45 µm syringe filter and transferred in a weighed 5 mL vial. The weight of the full vial was recorded immediately. The vials were placed in a fume hood for solvent evaporation. The remaining solid was dried in a vacuum oven at 45 °C for 5 h. All measurements were carried out in duplicate.

The solubility of pure BDMC $c_{\mathrm{sat},\mathrm{pure}\;\mathrm{BDMC}}$ was calculated from the ratio of the mass of the dried BDMC $m_{\mathrm{BDMC}}$ and the volume of the solvent $V_{\mathrm{solvent}}$:(1)$$c_{\mathrm{sat},\mathrm{pure}\;\mathrm{BDMC}\;}=\;\frac{m_{\mathrm{BDMC}}}{V_{\mathrm{solvent}}}{\;=\;m}_{\mathrm{BDMC}}\cdot \frac{\rho_{\mathrm{solvent}}}{m_{\mathrm{solvent}}},$$

The solubility of BDMC within the co-crystal $c_{\mathrm{sat},\mathrm{BDMC},\mathrm{cc}}$ was calculated by subtracting the concentration of HYQ in the solution $c_{\mathrm{HYQ}}$ from the total concentration $c_{\mathrm{total}}$ of the dissolved material. The concentration of HYQ was obtained from the initial mass of co-crystal $m_{\mathrm{cc}}^{0}$ added to the initial volume of solvent $V_{\mathrm{solvent}}^{0}$. Since HYQ has high solubility in EtOH and $H_{2}O$ [47], it was assumed that the whole initial amount of HYQ in the added co-crystal dissolved in the solvent. The concentration of dissolved material $c_{\mathrm{total}}$ was calculated from the ratio of the mass of the dried solid $m_{\mathrm{solid}}$ and the respective volume of solvent. $M_{\mathrm{HYQ}}$ and $M_{\mathrm{cc}}$ are the molar masses of HYQ and the co-crystal, respectively.
(2)$$c_{\mathrm{sat},\mathrm{BDMC},\mathrm{cc}\;}{=\;c}_{\mathrm{total}}\;-\;c_{\mathrm{HYQ}}\;=\;\frac{m_{\mathrm{solid}}}{V_{\mathrm{solvent}}}\;-\;\frac{m_{\mathrm{cc}}^{0}}{V_{\mathrm{solvent}}^{0}}\cdot \frac{M_{\mathrm{HYQ}}}{M_{\mathrm{cc}}},$$

### 3.8. Microscopic Analyses

Light microscopy was carried out on a Axioskop 2 microscope (Carl Zeiss, Oberkochen, Germany). Images were recorded and analyzed using Axiovision 2 software.

Scanning electron microscopy (SEM) was carried out on an XL 30 FEG (FEI, Hilsoro, OR, USA) at an acceleration voltage of 5 keV and a working distance of 12.4 mm. Powdered samples were adhered to a carbon patch and sputtered with platin to increase the conductivity of the sample. Secondary electrons and backscatter electrons were detected.

## 4. Conclusions

This study aimed to search for co-crystals of the curcuminoid BDMC with hydroxybenzenes. Two out of four hydroxybenzenes—pyrogallol (PYR), and hydroxyquinol (HYQ)—were found to form co-crystals with BDMC. Four different methods for co-crystallization were investigated. Co-crystallization from the eutectic melt turned out to be highly effective for the detection of a co-crystal phase with PYR where other methods were not successful due to its probable metastable state. Liquid-assisted grinding is a feasible method to produce a thermodynamically stable co-crystal with HYQ. The co-crystal composition of both co-crystals could be confirmed to be equimolar by DSC measurements of various molar compositions.

FTIR measurements provided structural information of the co-crystals since no suitable single crystals for single crystal XRD could be obtained. BDMC in the PYR-BDMC co-crystal is present in the diketo form, which is not the thermodynamically favored form of crystalline BDMC. In the HYQ-BDMC co-crystal BDMC exists in the keto-enol form. Spectral shifts to lower frequencies of various functional groups indicate a strengthening in intermolecular interactions within these co-crystals compared to its single components.

The solubility of BDMC in an ethanol/water mixture (50/50 *v*/*v*) could be notably increased by over 100% through co-crystallization with HYQ. Interestingly, the solubility of BDMC in pure ethanol slightly decreased through co-crystallization, which can be useful for capturing the target compound BDMC out of a solution containing the other curcuminoids. As CUR also forms co-crystals with HYQ, solubility measurements and construction of the ternary phase diagram might be helpful in order to find the optimum conditions for separating the curcuminoids by co-crystallization. Consequently, also further investigations on co-crystal formation of DMC are required. If DMC does not form co-crystals with HYQ, for example, this co-former can be used for the purification of DMC by removing CUR and BDMC from a curcuminoid mixture through co-crystallization. RES showed no affinity to form co-crystals with BDMC within this study but CUR did, giving rise to potential co-crystallization-based isolation of CUR. However, also other crystallization strategies might apply here and have to be considered. In summary, the results presented form a base for prospective research on curcuminoids separation by crystallization-based processes comprising co-crystallization.

## Figures and Tables

**Figure 1 molecules-26-00720-f001:**
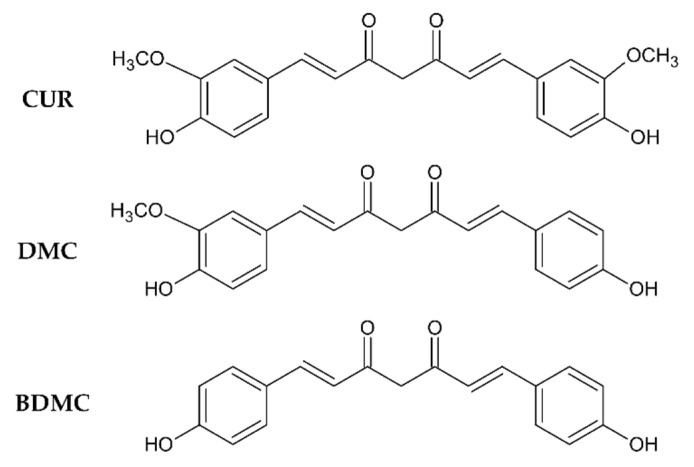
Molecular structures of curcumin (CUR), demethoxycurcumin (DMC), and bis(demethoxy)-curcumin (BDMC).

**Figure 2 molecules-26-00720-f002:**
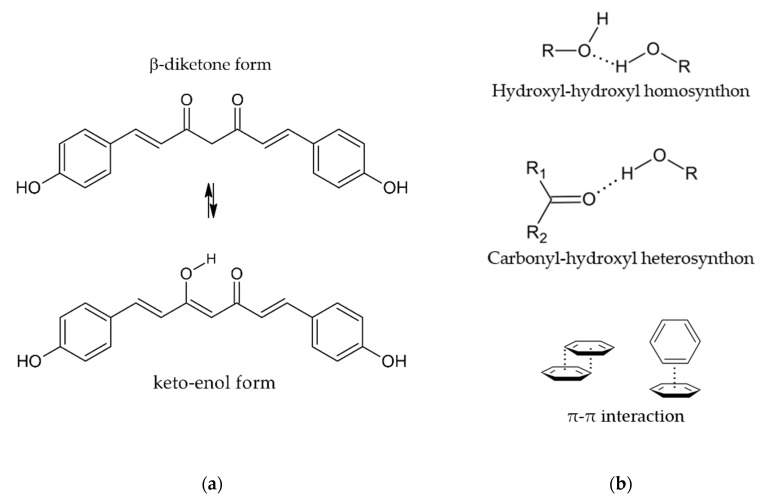
(**a**) Keto-enol tautomerism of BDMC, (**b**) Supramolecular synthons identified for the co-crystallization of BDMC.

**Figure 3 molecules-26-00720-f003:**
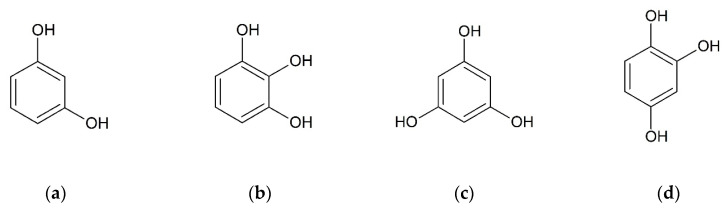
Chemical structures of selected coformers, (**a**) resorcinol (RES), (**b**) pyrogallol (PYR), (**c**) phloroglucinol (PHLO), (**d**) hydroxyquinol (HYQ).

**Figure 4 molecules-26-00720-f004:**
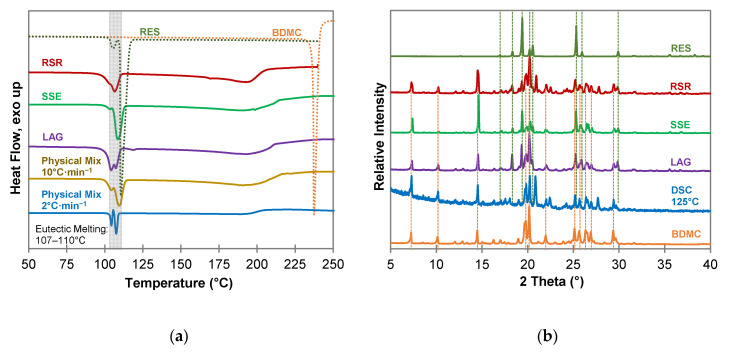
Characterization of BDMC, RES, and co-crystallization products of BDMC and RES, (**a**) DSC thermograms at a heating rate of 10 °C·min^−1^ and 2 °C·min^−1^ (blue line): DSC curves of pure substances are dotted, grey bar indicates eutectic melting; (**b**) PXRD diffractograms: characteristic peaks of pure RES and BDMC are marked as dotted green and orange line; abbreviations: RSR-rapid solvent removal, SSE-slow solvent evaporation, LAG-liquid assisted grinding, DSC-differential scanning calorimetry.

**Figure 5 molecules-26-00720-f005:**
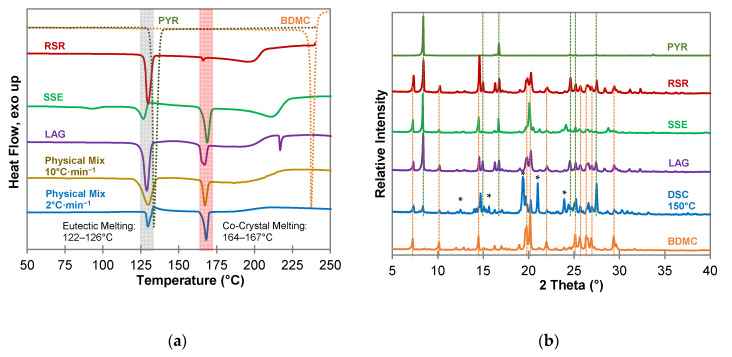
Characterization of BDMC, PYR, and co-crystallization products of BDMC and PYR, (**a**) DSC thermograms at a heating rate of 10 °C·min^−1^ and 2 °C·min^−1^ (blue line): DSC curves of pure substances are shown as dotted lines, grey bar indicates eutectic melting, red bar indicates co-crystal melting; (**b**) PXRD diffractograms: characteristic peaks of pure PYR and BDMC are marked as dotted green and orange lines, * new diffraction peaks; abbreviations: RSR-rapid solvent removal, SSE-slow solvent evaporation, LAG-liquid assisted grinding, DSC-differential scanning calorimetry.

**Figure 6 molecules-26-00720-f006:**
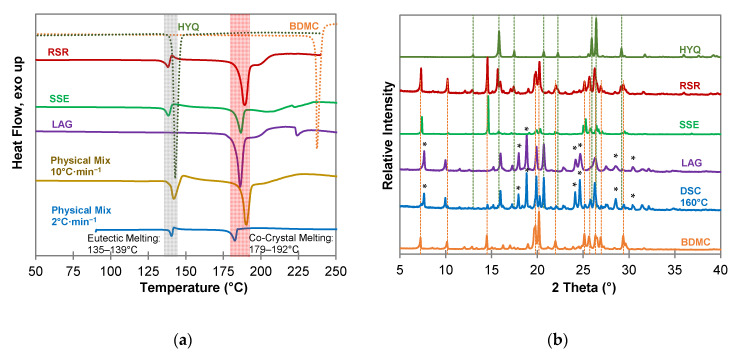
Characterization of BDMC, HYQ, and co-crystallization products of BDMC and HYQ, (**a**) DSC thermograms at a heating rate of 10 °C·min^−1^ and 2 °C·min^−1^ (blue line): DSC curves of pure substances are shown as dotted lines, grey bar indicates eutectic melting, red bar indicates co-crystal melting; (**b**) PXRD diffractograms: characteristic peaks of pure HYQ and BDMC are marked as dotted green and orange lines, * new diffraction peaks; abbreviations: RSR-rapid solvent removal, SSE-slow solvent evaporation, LAG-liquid assisted grinding, DSC-differential scanning calorimetry.

**Figure 7 molecules-26-00720-f007:**
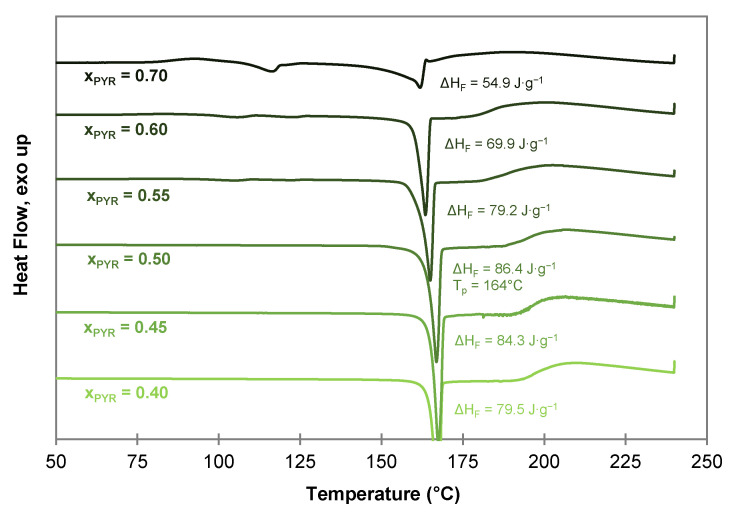
DSC thermograms of PYR-BDMC co-crystals produced by DSC at 150 °C with different initial compositions of PYR and BDMC at a heating rate of 2 °C·min^−1^.

**Figure 8 molecules-26-00720-f008:**
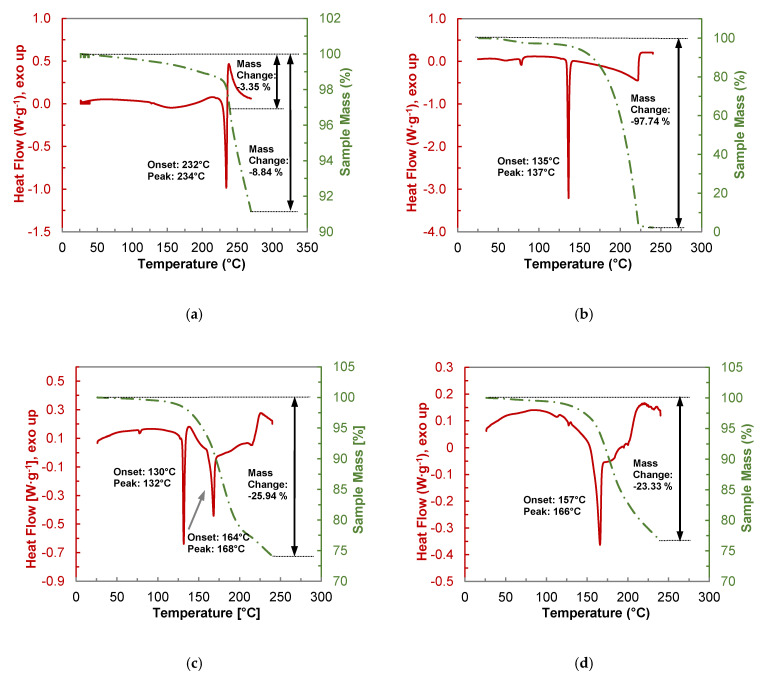
Thermogravimetric analysis (TGA-DSC) of (**a**) pure BDMC, (**b**) pure PYR, (**c**) a 1:1 physical mixture of BDMC and PYR, (**d**) BDMC-PYR co-crystal produced via DSC at 150 °C; heating rate: 2 °C·min^−1^.

**Figure 9 molecules-26-00720-f009:**
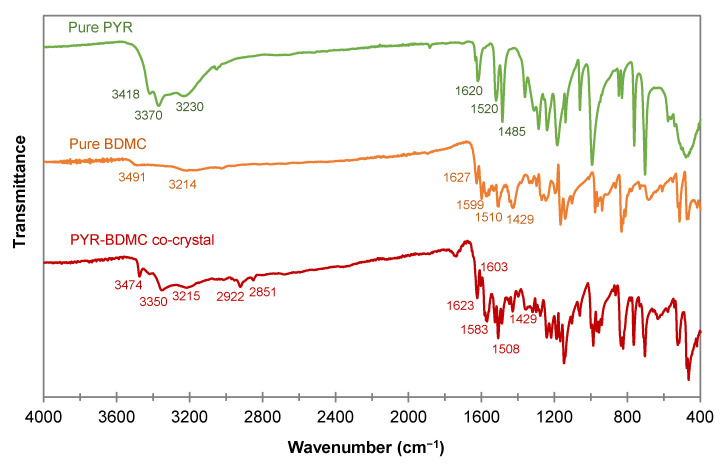
FTIR spectra of the PYR-BDMC co-crystal and its single constituents PYR and BDMC.

**Figure 10 molecules-26-00720-f010:**
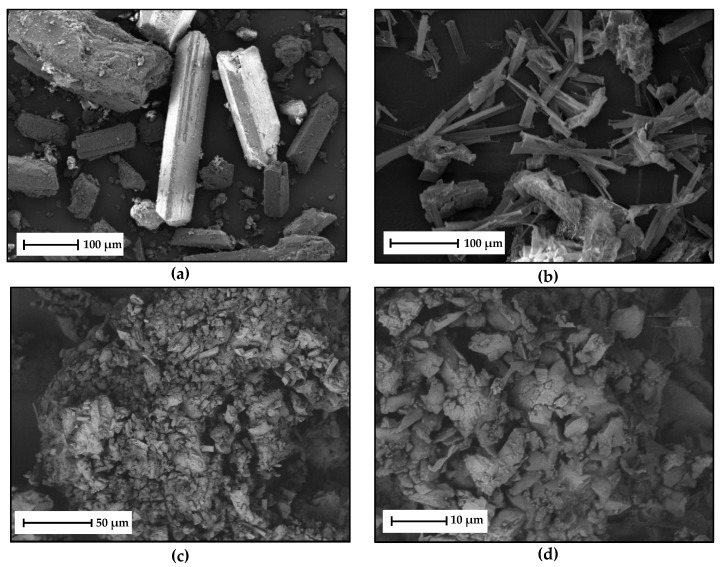
SEM micrographs of (**a**) pure BDMC crystals at 200× magnification, (**b**) pure PYR crystals at 250× magnification, (**c**) PYR-BDMC co-crystals produced via DSC at 500× magnification, (**d**) PYR-BDMC co-crystals produced via DSC at 2000× magnification; (**a**,**b**) detection of secondary electrons; (**c**,**d**) detection of back scatter electrons.

**Figure 11 molecules-26-00720-f011:**
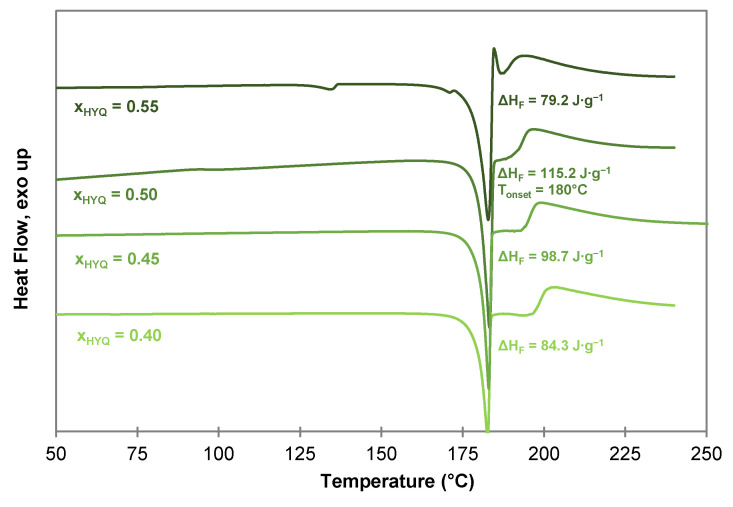
DSC thermograms of HYQ-BDMC co-crystals produced by LAG with different compositions of HYQ and BDMC at a heating rate of 2 °C·min^−1^.

**Figure 12 molecules-26-00720-f012:**
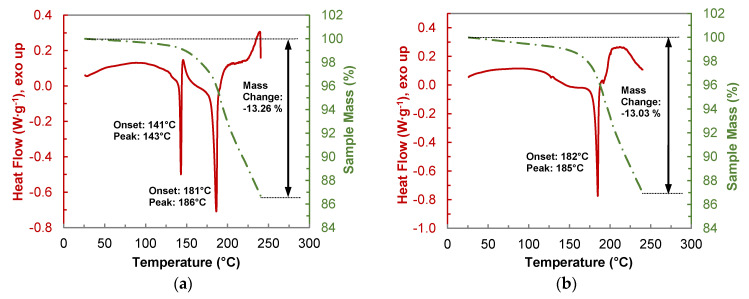
Thermogravimetric analysis (TGA-DSC) of (**a**) a 1:1 physical mixture of BDMC and HYQ, (**b**) BDMC-HYQ co-crystal produced by LAG; heating rate: 2 °C·min^−1^.

**Figure 13 molecules-26-00720-f013:**
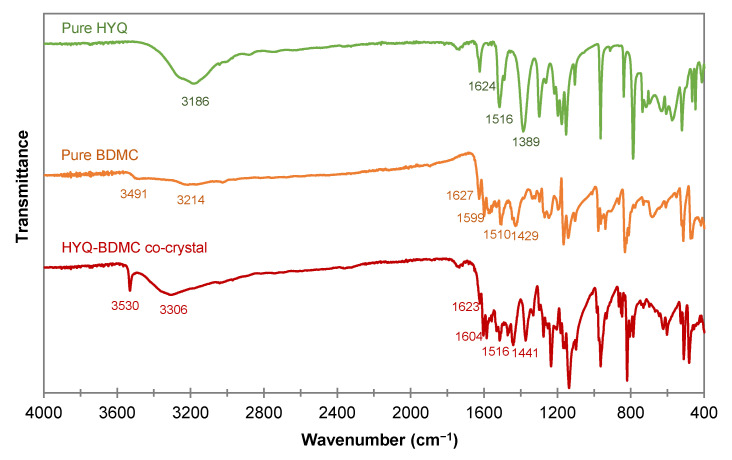
FTIR spectra of the HYQ-BDMC co-crystal and its single constituents HYQ, and BDMC.

**Figure 14 molecules-26-00720-f014:**
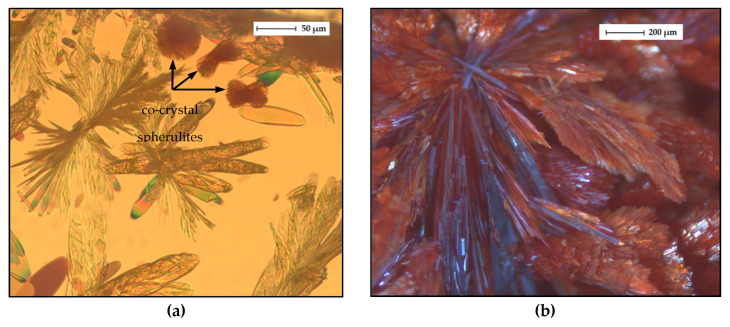
Light microscope images of (**a**) HYQ-BDMC co-crystals grown in an ethyl acetate solution, (**b**) a section of a HYQ-BDMC co-crystal grown from toluene/ethyl acetate (50/50 *v*/*v*).

**Figure 15 molecules-26-00720-f015:**
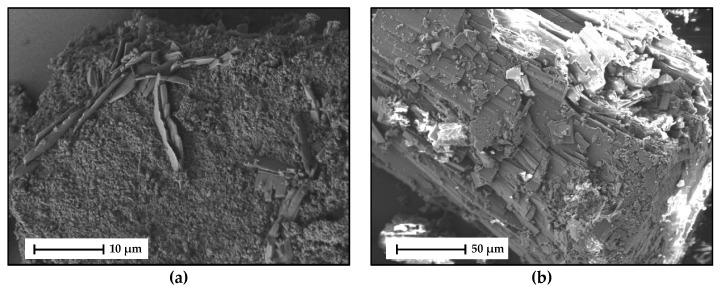
SEM micrographs of (**a**) HYQ-BDMC co-crystals produced by LAG at 2500× magnification, (**b**) HYQ-BDMC co-crystals produced by slow solvent evaporation from toluene/ethyl acetate (50/50 *v*/*v*) at 500× magnification; (**a**) detection of back scatter electrons; (**b**) detection of secondary electrons.

**Table 1 molecules-26-00720-t001:** FTIR vibration modes for the PYR-BDMC co-crystal and its single constituents PYR and BDMC, N.A.-Not Applicable.

	PYR	BDMC	PYR-BDMC Co-Crystal
Phenolic OH stretching (cm^−1^)	3418, 3370, 3230	3491, 3214	3474, 3350, 3215
C=O stretching (cm^−1^)	N.A.	1627	1623
Aromatic C=C stretching (cm^−1^)	1620	1599	1603
In-plane bending of enol C-O (cm^−1^)	N.A.	1429	1429
Methylene C-H stretching (cm^−1^)	N.A.	N.A.	2922, 2851

**Table 2 molecules-26-00720-t002:** FTIR vibration modes for the HYQ-BDMC co-crystal and its single constituents HYQ, and BDMC, N.A.-Not Applicable.

	HYQ	BDMC	HYQ-BDMC Co-Crystal
Phenolic OH stretching (cm^−1^)	3186	3491, 3214	3530, 3306
C=O stretching (cm^−1^)	N.A.	1627	1623
Aromatic C=C stretching (cm^−1^)	1624	1599	1604
In-plane bending of enol C-O (cm^−1^)	N.A.	1429	1441

**Table 3 molecules-26-00720-t003:** Solubility data for pure BDMC and BDMC within the HYQ-BDMC co-crystal in EtOH and EtOH/H_2_O (50/50 *v*/*v*) at 25 °C ± 1 °C.

	c_sat_ in EtOH (g·L^−1^)	c_sat_ in EtOH/H_2_O (50/50 *v*/*v*) (g·L^−1^)
Pure BDMC	43.25 ± 0.58	0.38 ± 0.05
BDMC within co-crystal	39.04 ± 1.10	0.83 ± 0.06

## Data Availability

The data presented in this study are available in manuscript and Appendix A.

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
