# Peer review of "A Contribution to the Solid State Forms of Bis(demethoxy)curcumin: Co-Crystal Screening and Characterization"

_molecules, 2021, doi:10.3390/molecules26030720_

Round 1

Reviewer 1 Report

The Manuscript with ID molecules-1062255, entitled “A Contribution to Solid State Forms of Bis(demethoxy)curcumin: Co-Crystals Screening and Characterization”, by Steffi Wünsche, Lina Yuan, Andreas Seidel-Morgenstern, Heike Lorenz, deals with the formation of co-crystals of Bis(demethoxy)curcumin (BDMC) with with pyrogallol (PYR), and hydroxyquinol (HYQ).  Various methods are applied to achieve co-crystallization, even though in the former case only crystallization from the melt gives the expected product, while in the latter case also liquid assisted grinding turns out to be effective. The obtained systems are tested/ characterized by powder X-ray diffraction, DSC, TGA, FTIR. Unfortunately, no single crystals X-ray measurements are possible, but the available data allow for the determination of the stoichiometric ratio in both co-crystal, and of the nature of the interaction between the two compounds in the lattice.

I think that the study is well conducted, with plenty of experimental data and details. The manuscript is well written and organized.  I recommend acceptance of the manuscript in the present form.

Reviewer 2 Report

Wünsche et al. investigated the co-crystals of the curcuminoid BDMC with hydroxybenzenes. Author obtained PYP and HYG, well described the overall process through experiments, and technical analysis was also performed very clearly. The flow and composition of manuscripts is very robust and provides useful experimental information. In particular, the detailed discussion of the experimental results in the manuscript is very valuable. I recommend the current manuscript to be published in the journal.

Minor.
- line 35-37: "The pharmacological benefits of curcumin making up 75-80 % of the curcuminoids are most extensively investigated." reference should be added.

- line 128: "For resorcinol (RES) (Figure 4), as well as for phloroglucinol (PHLO) (not shown here, see Figure S1), ....."
should be "For resorcinol (RES) (Figure 4), as well as for phloroglucinol (PHLO) (Figure S1), ...."

- In the manuscript, figures were indicated as "shown in Figure X" or "(Figure X)"in various ways. It is not obligatory, and it would be good to unify the picture notation for the reader's readability.

Reviewer 3 Report

Following are my concerns about this manuscript

  1. What is the main objective and application behind formation of BDMC cocrystals. Although the authors state that cocrystal formation might help with isolation, it was not demonstrated in the current manuscript. Either change the goal of the work or the benefit of isolation has to be demonstrated. Rest of the manuscript deals with mundane cocrystal screening and characterization using complementary techniques. 
  2. Is the selection of coformers exhaustive ? Is there a possibility of looking at additional coformers of cocrystal formation ? 
  3. The other way of looking at the manuscript could be looking for cocrystals to improve physicochemical / processing ability of BDMC, which would require significant amount of work.  

Round 2

Reviewer 3 Report

All my concerns have been addressed. 

The manuscript can be deemed acceptable for publication.